# Highly Flexible and Photo-Activating Acryl-Polyurethane for 3D Steric Architectures

**DOI:** 10.3390/polym13060844

**Published:** 2021-03-10

**Authors:** Ji-Hong Bae, Jong Chan Won, Won Bin Lim, Ju Hong Lee, Jin Gyu Min, Si Woo Kim, Ji-Hyo Kim, PilHo Huh

**Affiliations:** Department of Polymer Science and Engineering, Pusan National University, Busan 609-735, Korea; jihong.bae@pusan.ac.kr (J.-H.B.); jcwon@pusan.ac.kr (J.C.W.); freewonbin@nate.com (W.B.L.); dlwnghd15@pusan.ac.kr (J.H.L.); jg_min0629@naver.com (J.G.M.); siwoo3072@pusan.ac.kr (S.W.K.); sefg1530@naver.com (J.-H.K.)

**Keywords:** acryl-polyurethane, photocurable resin, photopolymer, three-dimensional printing architectures, digital light processing

## Abstract

An acryl-functionalized polyurethane (PU) series was successfully synthesized using poly(tetramethylene ether) glycol-methylene diphenyl diisocyanate (PTMG-MDI) oligomer based on urethane methacrylates to control the flexibility of photo-cured 3D printing architectures. The mass ratio of acryl-urethane prepolymer: 1,4-butanediol (BD) chain-extender: diphenyl(2,4,6-trimethylbenzoyl) phosphine oxide (TPO) photoinitiator was 10:0.25:1. To produce suitably hard and precisely curved 3D architectures, the optimal UV absorbance and exposure energy of the acryl-PTMG-MDI resin were controlled precisely. Owing to the optimized viscosity of the acryl-PTMG-MDI resins, they could be printed readily by digital light processing (DLP) to form precisely curved 3D architectures after mixing with 1,6-hexanediol diacrylate (HDDA). The acryl-PTMG-MDI formulations showed much better flexural resolution than the neat resins. The printed 3D structure exhibited high surface hardness, good mechanical strength, and high elasticity for flexible applications in consumer/industrial and biomedical fields.

## 1. Introduction

Photo-curing 3D printing based on the digital light processing (DLP) technique has been used consistently to produce complex architectures without the need for additional tools and machines [1,2]. DLP printing is an advanced technology involved in the light-mediated conversion of a monomer or oligomer resin to a solid photopolymer object [3,4,5,6,7,8]. UV-curable resins are suitable for the manufacture of complex and high-precision 3D objects in many fields. UV resin is a photo-curing system containing monomers, oligomers, and a photoinitiator, whose properties can be manipulated by exposure to light at specific wavelengths [9]. The use of UV-curable resins can provide many advantages, not only a fast curing range of seconds or minutes and energy efficiency by reduced processing, but also high resolution in the micrometer range under more intricate structures of smaller prints. Zhou et al. synthesized a photo acid generator (PAG) with a bis[(diarylamino)styryl] benzene core and covalently attached aryl sulfonium moieties (BSB-S2) to construct complex 3D structures with submicrometer feature sizes [10]. As a polymeric material for inkjet 3D printing, Schmidt et al. described urethane acrylate-based resins with viscosities ranging from 10 to 16 mPa.s, at temperatures ranging from 70 to 90 °C [11]. Although these UV-curable systems have been greatly optimized in individual 3D printing technology, some limitations, such as the overall energy efficiency, flexural resolution, and the great ability to achieve the desired properties, have become future challenges for 3D progress [12,13,14,15].

The DLP method using UV curing is getting attention due to its elaboration and fast speed of output. However, there is a need for supplementing material limitations and manufacturing precision. In addition, there are limitations such as poor durability, slow production speed, size, and miniaturization. This paper proposes the optimal formulation of UV-curable polyurethane resin. The optimum compounding ratio of photoactive acrylic urethane synthesis was derived for precision 3D printing materials and its properties were controlled using multifunctional monomer and photoinitiator. Using 3D printers using photocurable materials such as DLP and SLA (Stereo Lithography Apparatus) method, we intend to develop products with physical, mechanical, and chemical stability. The quality and performance of the resin in producing an object for DLP 3D printing were compared with those of a commercial counterpart.

## 2. Material and Methods

Scheme 1 presents a schematic diagram of the procedure to synthesize the acryl-PTMG-MDI resin. The molar ratio of HDDA and TPO was optimized precisely to achieve the proper flexural resolution of acryl-PTMG-MDI resin, as shown in Scheme 2. The characteristics of the UV-curable acryl-PTMG-MDI resin for DLP printing ability were evaluated in terms of the photoinitiator, monomers, and oligomers based on the viscosity and curing rate. UV-curable acryl-polyurethane (APU) was synthesized using three-step reactions. In the first step, poly(tetramethylene ether) glycol (PTMG, Mn = 1000 g/mol, Merck KGaA, Darmstadt, Germany) as a polyol and 4,4′-methylene bis(phenylisocyanate) (MDI, Tokyo Chemical Industry Co., Ltd., Tokyo, Japan) as a first diisocyanate were stirred mechanically to form the prepolymer for two hours at 50 °C with nitrogen charging. A 1,4-butanediol (1,4-BD, Tokyo Chemical Industry Co., Ltd., Tokyo, Japan) chain extender and second MDI in 5ml DMF (Duksan Chemical Co., Ltd., Seoul, Korea) were reacted sequentially to prepare the soft/hard backbone structure of polyurethane for one hour under 55 °C. Pentaerythritol triacrylate (PETA, Miwon Chemicals Co., Ltd., Ulsan, Korea) was chosen to produce the acylate multiple chain builder. 1,6-Hexanediol diacrylate (HDDA, Miwon Chemicals Co., Ltd., Ulsan, Korea) was used as the APU-based crosslinker. 1 wt.% diphenyl(2,4,6-trimethylbenzoyl) phosphine oxide (TPO, Merck KGaA, Darmstadt, Germany) was added to the reaction mixture under a stirring speed of 600 rpm at 30 °C, as photoinitiator for optimizing the UV-curing. To guarantee the 3D printing flexural resolution, commercially available UV-curable molecular structures that are suitable for the optimal UV absorbance-related DLP (IM2, Carima Co., Ltd., Seoul, Korea) 3D printing technique were designed, and their absorbing abilities were evaluated upon exposure to 385–405 nm UV radiation.

## 3. Results

Figure 1 shows the ability of the monomers, prepolymer, and APU to absorb at different wavelengths of the UV spectrum. The fundamentals of the UV absorbing ability in the constituents used have attracted little research attention, and almost no attempts have been made to optimize the absorption of the photopolymer in the UV band.

Figure 1a,c,d,f shows little absorption in the UV spectrum of 385–405 nm. The narrow regions of Figure 1b,g exhibited weak, localized photosensitivity after exposure to a 385–405 nm UV. The absorbing ability of the component shown in Figure 1e was directly proportional to the UV exposure range, and APU, including all components, also depended upon the exposure to UV radiation. This suggests that the combination of photoinitiator and acryl-urethane resin would allow strong UV absorbance, and the entire layer of the 3D architecture would be produced sufficiently through the individual exposure step.

DLP has some limitations in that the processing window of the UV-curable resin could be very narrow, which sets strict requirements regarding viscosity and resolution [16]. The formulation of UV-curable APU may be a key challenge because the maintenance of low viscosity can be essential for optimizing the balance between the components and processing [17]. Figure 2 shows the effect of adding APU on the viscosity of the tuning UV-curable resins using a Brookfield digital viscometer at 25 °C. The viscosity of the UV-curable resins was increased slightly from 100 mPa·s to 400 mPa·s by adding the weight fraction of APU ranging from 0 wt.% to 15 wt.%. The relative 3D printing ability of the 10 wt.% resin was higher than that of 15 wt.% because of the shorter waiting time for each layer.

Figure 3 shows the mechanical properties to examine the effects of added APU content on the flexural resolution of the 3.2 mm thick 3D-printed rectangular solid elastomeric architectures, which formed under the same printing condition. The increased addition of the APU content led to 8.3% to 9.37% improvements in flexural strength. As the layer-by-layer curing time could be dependent on the resin formulation, photoinitiator concentration, and curing conditions, the curing speed was also strongly dependent on the APU content, in the range of 0 wt.% to 15 wt.%. The addition of 10 wt.% APU may be the optimizing viscosity of UV-curable resin formulation because the 3D printing of this resin was carried out well within 10 s curing time for layer-by-layer growth. In particular, a 3D printed elastic architecture with 10 wt.% APU showed great flexural strength and good hardness within the limits of 3D printing conditions, such as a curing speed of 10 s^−1^. In particular, the flexural strength related to the lateral resolution was 3.7 MPa, according to tensile testing. The results could provide a challenging strategy to tune the flexural performance of the UV-curable resins by adding various APU contents. When the viscosity and curing time were 150 mPa∙s and 10 s, respectively, the UV-curable resin containing 10 wt.% APU was found to be a suitable candidate material for DLP-based 3D printing. The addition of APU improved the flexural strength, hardness, and viscosity of the UV-curable resins. The hardness and flexural strength value of the Carima Co. (Seoul, Korea) Ref. are 91.0 and 72.5, respectively. These results suggest that the APU content could be selected for a specific application or 3D processing based upon its effect on the viscosity, curing speed, architecture properties, and lateral resolution.

The mask-like architecture of a complex-shaped prototype was printed to confirm the quality of the flexural resolution. As shown in Figure 4, a 3D mask construction was DLP printed using APU-based resin during the individual layer-by-layer fabricating step. The uncured APU-based resin was removed using acetone or propylene glycol methyl ether acetate as a solvent to clean the surface of the printed 3D mask (Figure 4b). The 3D mask structure that corresponded well to the complex digital design observed, as shown in Figure 4a,c. The optimizing APU-based resin formulation for DLP printing may provide an opportunity to enable the relatively fast and flexural controlling construction of various architectures without any mold or machining. The interface resolution of the resulting 3D object may be achieved by balancing viscosity and energy optimizing to discriminately UV-curable APU-based resin formulation [18]. That is, the precision control of layer-by-layer may utilize components such as APU content and exposure energy.

As shown in Figure 5a–c, photographic images of the internal layer-by-layer portion, with exposure times ranging from 3.5 to 10 s, revealed a high-resolution pattern and a fine interface finish. Scanning electron microscopy (SEM) (Figure 5d–f) showed that this APU-based resin might allow the 3D printing of complex patterns with a high-resolution of approximately 5 μm. Depending on the flow viscosity and the curing speed of the APU-based resin, the solidification and interface resolution of each layer can be controlled precisely. During the 3D printing process, this UV-curable 10 wt.% APU-based system suffered few limitations, such as poor adhesion between layers, deformations, cracks, or porous appearance related to the viscosity and UV-absorbance. As shown in Figure 5d–f, the interface resolution in a building volume of 200 mm × 250 mm × 80 mm showed a much lower toughness and better accuracy between micro layer-by-layer structures. SEM (Figure 5d–f) showed that better surface adhesion of each layer was more important for providing a foundation for building a deliberate 3D object due to the hyposphere of the interface boundary. The distance of the interface was only about several hundred nm when the two layers were in contact, which may ensure a printed 3D structure with sufficient flexural strength without the need for additional processes. The printed 3D object was also equal to the layer height in between the built structure or the last completed layer. As shown in Figure 5e,f, the newly formed layers might have the same orientation as the built structure moves up. This phenomenon can be explained by the viscosity, build volume, speed, and workflow linked to the light-mediated conversion of UV-curable resin containing 10 wt.% APU oligomer to a solid 3D object. Moreover, an identical direction might act as a key solution to developing a complex 3D functional flexural architecture with a high degree of freedom. As a result, the finished 3D object of a UV-curable 10 wt.% APU-based formulation may have a suitable hardness and flexural-strength.

## 4. Conclusions

The formulation of a novel UV-curable APU-based resin was applied to the DLP printing process. The direct application of a 10 wt.% APU-based formulation in normal DLP printing was proven by the successful formation of a complex 3D mask architecture with good hardness and excellent flexural-strength. The DLP printable 10 wt.% APU-based resin was obtained by the optimal mixing of HDDA and APU at various ratios. This resin provided a 3D elastic object that could reach a hardness and flexural strength of up to 91 and 36.8 MPa, respectively, which were higher than those of the normal UV-curable elastomers. Therefore, the 10 wt.% APU-based formulation that can be photopolymerized by strongly absorbing irradiation might lead to a complex shaped architecture with high flexural-resolution and excellent surface finish. The APU-based resin system could be a promising candidate for producing flexible and reversible 3D structures in many applications for industries, such as soft robots, flexible electronics, and smart materials [19].

## Data Availability

The data presented in this study are available on request from the corresponding author.

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
