# Peer review of "Highly Flexible and Photo-Activating Acryl-Polyurethane for 3D Steric Architectures"

_polymers, 2021, doi:10.3390/polym13060844_

Round 1

Reviewer 1 Report

The authors have demonstrated capability to obtain DLP printing through UV-curable APU-based resin.I recommend this manuscript to be accepted after some minor revision. Hopefully, this approach will open many potential applications of 3D printing in the high precision submicron domain and will also overcome the previously reported challenges.

The authors do not list where they obtained the diphenyl(2,4,6-tri- 66 methylbenzoyl) phosphine oxide (TPO) initiator used in this work (please provide commercial sources).

Could you provide information on the ratio of inhibitor to initiator will drastically effect gelation and polymerization rates?

the initiator in this work, TPO, should be listed in the experimental section. Currently it can only be found in the SI, the experimental section simply says “initiator”.

The thickness of the base layer will surely be affected by the writing parameters as much as the other dimensions. Did the authors account for this?

Reviewer 2 Report

This is good, interesting work. It is well written and easy to read/understand.

I have a few comments to the authors that I assume might help to improve the manuscript.

Line 13: "critical point" - this is not entirely clear. It is not mentioning anywhere later in the text.

Line 31 (and 39): "The use of UV-curable resins can provide ...energy efficiency..." not clear what do you mean here by "energy efficiency". Potential energy for resin curing? UV source energy? Please clarify.

Can you define the role of each chemical (monomer, initiator..)?

Figure 1 legend is not convenient to refer to the text every time, I would suggest using the name of the materials instead of lettering. What is the logic to separate the figures? Are the materials (a)–(d) and (f)–(h) are grouped for some reason?

Figure 1 caption: don't use full name and suppliers.

Line 51–52. I did not found the comparison with commercial counterpart. Please clarify.

Figure 3 is redundant because Table 1 has the same data.

You may want to define what is "curing speed" and "exposure time" (and their relation).

Line 117-118: Did you run the experiments on the influence of APU on the curing speed? 10 to 1000 1/s seems very significant range.

 156-158: After this sentence, it is expected you say how exactly interface resolution can be controlled?

Overall, can you benchmark your material against commercially available? 
